

# Constantly renewing glacial lakes in the Kyrgyz Range, northern Tien Shan

Mirlan Daiyrov[1], Chiyuki Narama[2]

[1]Graduate School of Science and Technology, Niigata University, Niigata 950-2181, Japan
[2]Program of Field Research in the Environmental Sciences, Niigata University, Niigata 950-2181, Japan

*Correspondence to*: Mirlan Daiyrov (mirlan085@gmail.com)

**Abstract.** Glacial lakes in the Kyrgyz Range of the northern Tien Shan, Kyrgyz Republic, Central Asia are monitored due to concern over possible glacial lake outburst floods (GLOFs) after recent glacier shrinkage in the area. To evaluate the status of these lakes, we investigated the number of glacial lakes and the lake area from 1968 to 2021 using Corona KH-4, Landsat (7 and 8), Sentinel-2, and PlanetScope satellite images. We found that the number of glacial lakes increased by 30% from 417 in 1968 to 543 after 2000. However, some lakes vanished. In particular, 305 of the original 417 glacial lakes vanished by 2000, whereas 431 of the 543 glacial lakes identified between 2000 and 2021 formed after 2000. In addition, the total lake area significantly increased from 0.87 km$^2$ in 1968 to 5.21 km$^2$ in 2021. New glacial lakes rapidly formed because the glacier area has decreased by 32% over the past 50 years, and the places from which glaciers retreated have changed into glacier-moraine complexes (GMCs) on which new lake basins appeared and expanded. Thus, the renewal of glacial lakes in the Kyrgyz Range is the outcome of glacier shrinkage and ice melting within GMCs.

## 1 Introduction

In the Kyrgyz Range in the northern Tien Shan, many glacial lakes have been identified using satellite data (Kattel et al., 2020; Daiyrov et al., 2022). In the region, the appearance of new glacial lakes is presumably due to recent glacier shrinkage under global warming (Aizen et al., 2006; Bolch et al., 2015). Such lakes should be monitored because they often cause glacial lake outburst floods (GLOFs) (Erokhin et al., 2008, 2017; Narama et al., 2010, 2018; Kattel et al., 2020), which could affect the millions of people who live downstream.

Indeed, several GLOFs have occurred in the Kyrgyz Range within the past 20 years. On 5 June 2009, the Noruz Valley experienced catastrophic debris flow from the Takyrtor glacial lake (Kattel et al., 2020). Then, on 31 July 2012, the Adygene River valley was affected by drainage from the Teztor glacial lake (Erokhin et al., 2017). Five years later, on 12 August 2017, the Noruz River basin flooded when the Chelektor glacial lake experienced drainage (Daiyrov et al., 2022). The most recent flood, on 2 August 2021, originated from the Akpai glacial lake in the Sokuluk River basin. The above GLOFs damaged roads, bridges, and agricultural fields.





The glacial lakes in the northern Tien Shan tend to be much smaller than those in the eastern Himalayas (Yamada et al., 1998, Komori et al., 2004; Nagai et al., 2017). Additionally, these lakes experience unstable fluctuations and seasonal area changes (Daiyrov et al., 2018; 2022), in contrast to the Himalayan lakes which have been expanding since the 1950s. Local geomorphological conditions, specifically ice tunnels in glacier-moraine complexes (GMCs), are the main factors affecting seasonal lake changes (Daiyrov et al., 2018; Narama et al., 2018; Daiyrov and Narama, 2021). Such unstable lakes, regarded

as short-lived (Narama et al., 2010, 2018) or non-stationary (Erokhin et al., 2017) lakes, cause GLOFs in this region.

To better understand how these glacial lakes develop, we analysed satellite data for the numbers and areas of the lakes, as well as changes in glaciers and GMCs, during 1968–2021. The early satellite images from Corona KH-4 in 1968 are compared to optical satellite images from Landsat (7 and 8) for 2000–2013, to Sentinel-2 for 2014–2021, and to PlanetScope for 2017–2021 to assess historical as well as recent variations in glacial lakes. We also examined surface changes of GMCs

based on digital surface models (DSMs) from Corona KH-4 (1968), AeroPhoto (1991), ALOS/PRISM (2010), and the High Mountain Asia 8-m Digital Elevation Model (DEM) derived from DigitalGlobe satellites (2011, 2013).

## 2 Study area

The study area lies within Kyrgyzstan in the Kyrgyz Range of the northern Tien Shan (Fig. 1). The mountain ridges of this range are from 2500 to 4900 m above sea level, and along the northern flank in the central part (4200–4500 m) of the range

lie many glaciers at higher elevation than those in the eastern and western parts. Distributed at their fronts are GMCs, landforms consisting of much dead ice and debris (Shatravin, 2007) that form when the glacier retreats gradually. Some GMCs have changed into glacier-derived rock glaciers, but those that remain can have small lakes such as a short-lived lake or a thermokarst lake (Narama et al., 2018). In recent decades, glaciers and GMCs have been affected by climate change, leading to their shrinkage and degradation (Daiyrov et al., 2018). Though many still contain significant ice, some GMCs in

the Kyrgyz Range are almost ice-free (Erokhin et al., 2017; Daiyrov et al., 2022).

In the Kyrgyz Range, 483 glaciers have been identified at 3100–4200 m asl, covering an area of 520 km2 (Usubaliev et al., 2013). According to Maksimov and Osmonov (1995), most of these glaciers are in the Issyk-Ata, Alamudun, West-Karakol, and Sokuluk River basins (Fig. 1). Bolch (2015) reported that the glacier area in the Ala-Archa Basin decreased by 18% from 1964 to 2010, consistent with the earlier finding from Aizen et al. (2006). The mass balance of the Ala-Archa River

basin was only reported for Golubin Glacier, the largest glacier in the Kyrgyz Range (5.42 km2). It averaged –0.20 ± 0.42 m w.e./year from 1949/1950 to 2020/2021 (Azizov et al., 2022). The largest GMC (approximately 3 km2) in the range is the Ken-Tor glacier front in the Noruz River basin (Maksimov and Osmonov, 1995).

Precipitation is an important factor affecting glacier mass balance during spring and summer (Ponomarenko, 1976; Aizen et al., 2006). The central part of the range receives the most precipitation, around 500–925 mm annually. The average monthly

temperature at 2500–4000 m varies from 1.88 to 12.38°C in July and from –15.48 to –9.4°C in January (Podrezov, 2013).



## 3 Methods

### 3.1 Corona image data

The early images from the Corona KH-4 satellite come from 13 photographic images of nearly cloud-free scenes of the
Kyrgyz Range taken in 1964 and 1968 to determine the areas and numbers of glacial lakes and glaciers. Only 14 glaciers on
the east side of the Kyrgyz Range are viewable in 1964, amounting to 2% of the study area, yet 98% of the study area is
included in the 1968 Corona images.

Each Corona image covers an area 16 km wide, with three consecutive images in a series, and an image resolutionof 1.8–
2.70 m (Table 1). To make Corona DSMs and orthophotos, we used stereo-pair images of forward and aft-looking image
pairs with Metashape software (Agisoft). The geometric distortion of the Corona images are corrected in a selected non-
metric camera model. Ground control points (GCPs) are identified for Corona data based on locations from Google Earth
Pro (accurate coordinates from QuickBird images) and ALOS PRISM images in 2007 and on elevations from DEMs of the
National Snow and Ice Data Center's (NSIDC) High Mountain Asia (HMA) images in 2010 and 2011. We use stable points
(e.g., large boulders) outside of glaciers and GMCs as the GCPs. For each image (backward and forward), we use 25–30
GCPs (dividing each scene into four groups: a, b, c, and d). Cloud-covered areas are excluded from the analysis. For the
analysis of the 1991 aerial images, the same methods are implemented. Aerial photography images are only used for the
GMC of the Jylamysh Glacier front to investigate surface changes.

### 3.2 Mapping of glacial lake and glacier

We manually map glacial lakes on both the Corona KH-4 (1964 and 1968) and recent optical satellite images, including
Landsat 7/ETM for 1990–2010, Landsat 8/OLI for 2010–2021, Sentinel-2 for 2016–2021, and PlanetScope for 2016–2021.
Corona images in 1964 and 1968 include shadows or snow cover regions around lakes, making it hard to determine if such
regions are over the corresponding lake. To assign such regions as lake, we examined the slope angle of the region, calling it
as lake if the slope is less than 10 degrees according to the Corona DSM. However, we exclude such regions from mapping
if we cannot distinguish the lake area due to shadow or snow cover. We use a 100-m2 threshold to extract lakes, in
accordance with the threshold from Daiyrov et al. (2022). Uncertainty is estimated using methods from Hanshaw and
Bookhagen (2014).

We also mapped glaciers using the Corona KH-4 (1964 and 1968) and PlanetScope for 2021. Area changes are calculated by
polygon data in 1964/1968 and 2021.

### 3.3 Geomorphologic analysis of basins on GMC

We examine the GMCs in the study area, including ground ice based on fringe areas (motion areas), using a differential
interferometric SAR (DInSAR) analysis using GAMMA SAR software and ALOS/PALSAR 1-2 data. We use 49 image
pairs from ALOS-2/PALSAR (n = 18) and PALSAR-2 (n = 31) data between 2009–2010 and 2014–2022. For details of the

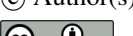



method, see previous reports (e.g., Goldstein et al., 1997; Werner et al., 2000; Quincey et al., 2007; Sandwell et al., 2008; Daiyrov et al., 2018).

To research the surface changes of lake basins in the central part of the Kyrgyz Range, we use DEMs generated from Corona KH-4B images (1968), aerophotographic imagery (1991), and ALOS/PRISM data (2010). Lake basins in 1968, 1991, 2010, and 2013 are extracted using a filling model in ArcGIS Pro. The 1991 data include only the Jylamysh Glacier front area.

## 4 Results

### 4.1 Numbers and areas of glacial lakes during 1968–2021

Results in Fig. 2 show 417 glacial lakes in 1968 and 543 in 2000–2021, an increase of 30%. However, by 2021, 305 (73%) of the original 417 lakes vanished while 431 new lakes appeared. 112 lakes had survived from 1968 to 2021. Compared to the glacial lakes in the eastern Himalayas that tend to persist and grow since the 1950s (Yamada, 1998), the glacial lakes here are more transient, similar to the patterns in the Kungoy and Ili ranges (Narama et al., 2009).

Another difference from the glacial lakes of the eastern Himalayas (Nagai et al., 2017) is that these lakes are relatively small.

Consider the three size classes in Fig. 3. The number of small glacial lakes (0.0001–0.001 km2) comprise 56% of all lakes in 1968 and 47% of all lakes in 2000–2021. However, over this period, the trends indicate an increase in the areas of the lakes, leading to a relative increase in the number of medium and large lakes. For instance, the number of large lakes (0.01–0.1 km2) increase from 13 in 1968 to 64 in 2000–2021. Although there are still many small lakes, the medium-sized lakes (0.001–0.01 km2) occupy 57% and 50% of the total lake area in 1968 and 2000–2021, respectively. The total area of glacial

lakes increased sixfold from 0.87 km2 in 1968 to 5.21 km2 in 2000–2021. Notably, the total area of lakes in the middle size range (0.001–0.01 km2) increased 5.2-fold between 1968 and 2000–2021. Of all the lakes that survived from 1968 to 2021, 112 (27%) showed a large change in their area. Among these 112 lakes, we confirmed that 16 were unstable, cumulatively increasing in area from 0.005 to 0.053 km2.

### 4.2 Types of glacial lakes and their change

In this region, we distinguish the glacial lakes in contact with a glacier from those that do not contact (Fig. 4a). In 1968, 61% (256 out of 417) of the lakes are the contactless type on GMCs (Fig. 4b). After 1968, this percentage increases to 75% as the number of contact-type lakes slightly decrease to 2000–2021 and the number of contactless-type lakes substantially increase. Figure 4c shows the change in lake types from 1968 to 2000–2021. Many lakes in contact with glacier fronts in 1968 became contactless. In 1968, 161 lakes were the contact type (including one from 1964); of these, only 17 lakes remained unchanged,

34 lakes became contactless as the glacier retreated, and the remaining 110 lakes disappeared. Of the 256 contactless lakes in 1968 (including one from 1964), 58 remained and 198 disappeared. These results indicate that it is difficult for contact type lakes at a glacier terminus to continue expanding for a long period, unlike lakes in the eastern Himalayas (Yamada, 1998).


### 4.3 Area changes in glaciers and surface changes in GMCs

To better understand the reason for the recent formation and disappearance of glacial lakes, we investigated changes in the relevant glaciers and GMCs during this period. Glacier termini have substantially retreated since 1968, with an example shown in Fig. 5a. The total glacier area in the Kyrgyz Range was 390.3 km2 in 1968, but it had decreased to 265.4 km2 by 2021, representing glacier shrinkage by 32% over 53 years (Fig. 5b). The central part of the Kyrgyz Range contains 72% of the glacier area, whereas the western and eastern parts constitute 19% and 9%, respectively. The glacier shrinkage in the

central part was 30%, versus 36% and 40% in western and eastern parts, respectively.

Many contactless type lakes exist on GMCs, which also changed. Between 1968 and 2010/2013, the DSM data indicates that 112 (18%) of the 611 GMCs in the Kyrgyz Range experienced significant decline (from –5 to –30 m). Notably, lake basins with large surface declines (from –10 to –35m) were found to lie on GMCs in the Sokuluk, Jylamysh, Ala-Archa, Alamudun, Noruz, Issyk-Ata, and Kegeti River basins (Fig. 1). Furthermore, DInSAR data for 2007–2010 and 2014–2020 showed that

450 (74%) of the 611 GMCs underwent significant displacement, indicating surface changes due to melting ice (Fig. 5c).

On the GMC of Jylamysh glacier in the central part of the range, the numbers and sizes of lake basins continually increased through 1968, 1991, 2010, and 2021 (Fig. 6a-d). The DInSAR data (1 September 2009—20 July 2010) in Fig. 6e shows substantial surface motion on the GMC due to melting ground ice (Fig. 6e). New lake basins formed and expanded in these areas of displacement, indicating ice melting. In this GMC, the glacial lake area remains essentially constant between the

periods of 1968, 1991, 2010, and 2021, whereas the areas of lake basins in 2021 increased eightfold compared with 1968 (Fig. 6f). Meanwhile, the numbers of glacial lakes and lake basins increase from 1968 to 1991 and then decrease until 2021.

Consider the surface changes of two profiles in the GMCs of the Julamysh and Chelektor glaciers. For the Julamysh Glacier, Fig. 7a,b shows the area of a lake basin on the GMC has expanded significantly. Along a cross-section, a comparison of DEM differences (Fig. 6a) between 1991 and 2013 show a surface decline of up to –35 m (Fig. 7c), which is the largest

measured decline in the Kyrgyz Range. The corresponding data for Chelektor shows two new glacial lakes form and an increase in the lake basin are as the glacier retreated (Fig. 7d-f).

### 4.4 Differences in the glacial lake situation among river basins

We now consider all the individual river basins. For each basin in the region, we show the changing number of lakes, glacial area, and area of GMCs in Fig. 8a. The number of glacial lakes has increased in most river basins, but 79% of the total lakes

appeared after 2000 (Fig. 2). As of 2021, the basins with the most glacial lakes are Issyk-Ata (83), Sokuluk (53), Alamudun (51), and Ala-Archa (37) in the central part of the range, as well as the Shamshy-North (53) in the eastern part and Ak-Suu (35) in the western part. All other valleys have less than 30 lakes. Figure 8a also shows that the number of glacial lakes is strongly related to loss area of glaciers. For example, the Sokuluk, Ala-Archa, Alamudun and Issyk-Ata river basins have both a relatively large number of lakes and a relatively large loss of glacial area. These basins also have relatively large

GMCs, which is consistent with the glacial lakes tending to exist on GMCs.





We also confirmed the development of many lake basins in the Sokuluk, Jylamysh, Ala-Archa, Alamudun, Noruz River, and Issyk-Ata river basins in the central part of the range (Fig. 8b). The formation of lake basins due to surface changes also led to many new glacial lakes. Specifically, the Sokuluk, Ala-Archa, Alamudun, and Issyk-Ata river basins experienced the formation of many new lakes. This result is consistent with the area of GMCs (Fig. 8a).

## 5 Discussion

### 5.1 Why many glacial lakes rapidly renew

Of the region's extant glacial lakes, 79% are new lakes that appeared since 2000 (Fig. 2). Why? Consider that the situation here is similar to those in the Kungoy and Ili Ranges (Narama et al., 2009). Consistent with the glacier trends there, the glacial area here decreased by 30% over the past 53 years. And although the number of contact type lakes decreased from 1968 to 2000–2021, the contactless type lakes increased by an even larger number. This increase in contactless lakes is likely related to the glacier shrinkage after 1968 and the resulting expansion of the GMC area into the regions once glaciated (Fig. 5a). Additionally, ice melting within GMCs likely contributed to lake basin formation and expansion. For example, we confirmed an expansion of the banks of a lake depression at the GMC of the Jylamysh glacier front, which had occurred due to both downwasting and backwasting from the melting of underground ice (Figs. 6, 7). Such surface changes are similar to those reported for the debris-covered area of glaciers (Goldstein and Werner, 1997). Similarly, in the Teskey Range, basins are produced via downwasting of the glacier surface due to glacier retreat (Narama et al., 2010). As glaciers and dead ice melt, the meltwater fills a basin that eventually leads to lake formation.

Recent glacier shrinkage did not cause the expansion of terminal lakes, in contrast to the situation in the eastern Himalayas (Yamada, 1998; Ageta et al., 2000; Iwata et al., 2002; Komori et al., 2004; Nagai et al., 2017). Instead, the sudden glacier retreat caused the contact-type lakes to separate from their glacier and become contactless (Fig. 4c). As a result, glacier meltwater no longer directly flows into these lake basins, and these lakes have disappeared. Narama et al. (2018) also argued that the connection between glacier and lake via ice tunnels with water channels on the GMC is an important factor in the lake's formation. Thus, the lake's existence is mainly influenced by regional geomorphological conditions such as ice tunnels, the presence of ground ice, and the formation of depressions on GMCs. After these contactless lakes vanish, melting ground ice causes the creation of new lake basins or the expansion of old basins. When these basins contact a water channel or meltwater from dead ice pools as a thermokarst lake, new lakes appear within the basins. Some new lakes have reappeared where contactless types were previously located. These lakes are mainly related to the existence of ground ice, stream channels, and the lake-basin formation. The long-lived lakes survived by collecting channelled meltwater from glaciers since 1968. Reasons for losing a lake include having the river bed of the glacier front being inclined and having no clear moraine ridges on the GMC.

Recent climate change has affected the GMCs, which play an important role in the formation of glacial lakes. In recent years the internal structures of the GMCs became more sensitive to climate change, producing more rapid changes (Daiyrov et al.,





2018, 2020). As lake basins may be essential for future lake formation, GMCs are potential environments that can harbor an increasing number of contactless and new glacial lakes. Falatkova et al. (2019) noted that glacier mass loss led to glacial

lakes in the Kyrgyz Range. However, the current situation of glacial lakes there is likely the result of glacier shrinkage and ice melting inside GMCs. In contrast, the formation of glacial lakes in the Himalayas is strongly linked to glacier shrinkage. Himalayan glaciers have large glacier areas and moraines (ICIMOD/UNEP, 2001; Iwata et al., 2002), with glacial lakes forming there since the 1950s, indicating their long existence (Yamada, 1998). In the Kyrgyz Range, this situation might change under a warmer environment at higher altitudes due to changes in mountain permafrost (Marchenko et al., 2007) and

an increase in the minimum glacier elevation (Niederer et al., 2008).

## 5.2 Mitigating the danger from glacial lake outburst floods (GLOFs)

In the last 20 years, many GLOFs have occurred in the central part of the Kyrgyz Range (Erokhin et al., 2017; Kattel et al., 2020; Daiyrov et al., 2022). Here lie the majority of glacial lakes in the range, particularly in the Sokuluk, Ala-Archa, Alamudun, and Issyk-Ata River basins. Moreover, these basins have relatively large areas of GMCs that might be related to

large glacier shrinkage (Fig. 8a). Moreover, as shown by DInSAR analysis, the central part of the range has unstable GMCs with substantial ice potential, resulting in the emergence of numerous lakes and renewal of glacial lakes. Thus, the central part of the range has experienced large changes to its glacial lakes. The central part of the Kyrgyz Range is also home to most of the population in the Chui Region, and many people live downstream from these glacial lakes. In contrast, other river basins in the range have relatively few glacial lakes in ice-cored moraines and have glaciers with relatively little

shrinkage as well as relatively stable surfaces. Rising temperatures in the Kyrgyz Range can accelerate melting processes in GMCs , enlarge lake basins, and lead to the formation of new glacial lakes. Such changes will result in a future drastic renewal of many lakes and expansion of ice tunnels might be high for GLOF risk. Therefore, these areas should be continuously monitored.

## 6 Conclusion

This study identified 417 glacial lakes in 1968 and 543 glacial lakes in 2000–2021. Although the number of glacial lakes increased by 30%, 73% of the original lakes vanished (305), the other 112 lakes persisting through 2021. This loss was offset by the formation of 431 new lakes after 2000. We concluded that the new glacial lakes formed rapidly because the glacier area decreased by 32% from 390.3 to 265.4 km2 between 1968 and 2021. The regions of glacial retreat became GMCs. We identified 611 GMCs in the Kyrgyz Range, with most containing dead ice. Melting of this ice caused considerable surface

subsidence, forming many contactless type lakes and lake basins in the GMCs. Thus, the increasing area of GMCs can provide greater area for contactless lakes, which can stimulate growth in their number and total area. With such growth, the danger of glacial lake outburst floods may increase in the area.



**Author contributions**

MD and CN conducted the field survey and performed an analysis of field and satellite data. MD wrote the paper. CN improved the manuscript and suggested some discussion points. All authors read and agreed to the published version of the paper.

**Competing interests**

The authors declare that they have no conflict of interest.

**Funding**

Grant-in-Aid for Scientific Research (B) (19H01372 and 23K22023) of MEXT KAKENHI Grant.

**Acknowledgments**

This work was supported by Grant-in-Aid for JSPS Fellows (22F22006) of JSPS KAKENHI Grant and Grant-in-Aid for Scientific Research (B) (19H01372 and 23K22023) of MEXT KAKENHI Grant. This study used ALOS satellite image data from ALOS Research Announcement (RA) in the framework of JAXA EORC.


**Table 1: Corona KH-4 images of the study area (1964–1968).**

| Satellite | Corona Scenes | Date | Ground resolution (m) | Coverage area of study side and percentage |
|---|---|---|---|---|
| KH-4A 10 (Corona 85, Mission 1010, OPS 3497) | DS1010-2086DA121 DS1010-2086DA120 DS1010-2086DF114 DS1010-2086DF115 | 20.09.1964 | 2.70 | East part – 2% |
| KH-4A 48 (Corona 128, Mission 1048, OPS 0165) | DS1048-1039DA030 DS1048-1039DA031 DS1048-1039DA032 DS1048-1039DF030 DS1048-1039DF031 DS1048-1039DF032 | 18.09.1968 | 2.70-7.60 | West, central and east part – 74% |
| KH-4B 4 (Corona 127, Mission 1104, OPS 5955) | DS1104-2185DF053 DS1104-2185DA057 DS1104-2185DA058 | 07.08.1968 | 1.80 | Central-south part, East part – 24% |




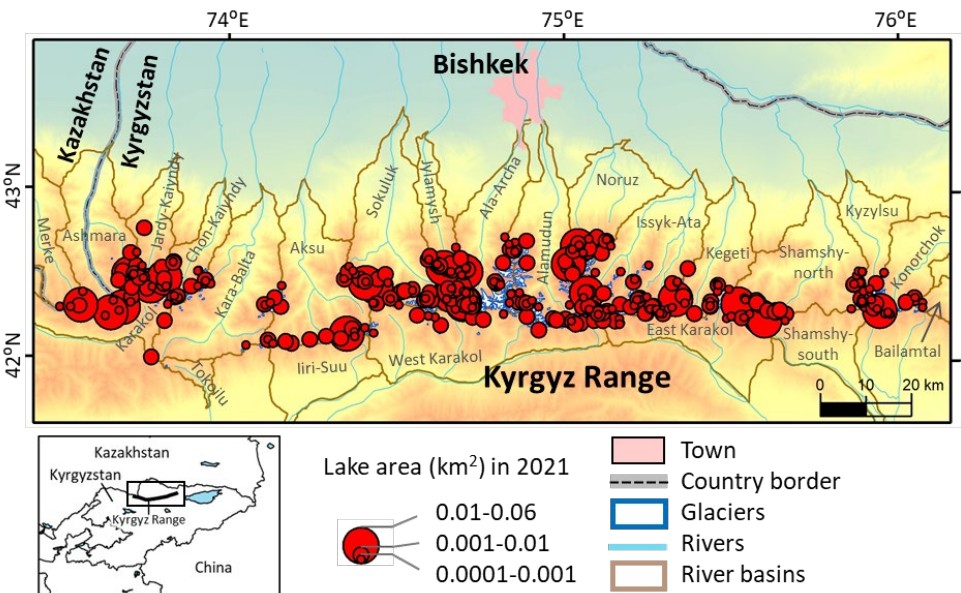

**Figure 1: The study area. Top shows the lake locations and areas in 2021 as well as other geographical features. Bottom left shows the location of the study area.**

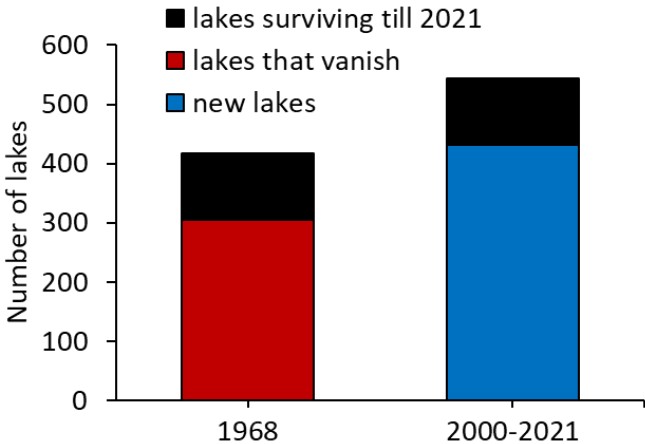

**Figure 2: Number of glacial lakes in 1968 and 2000–2021.**




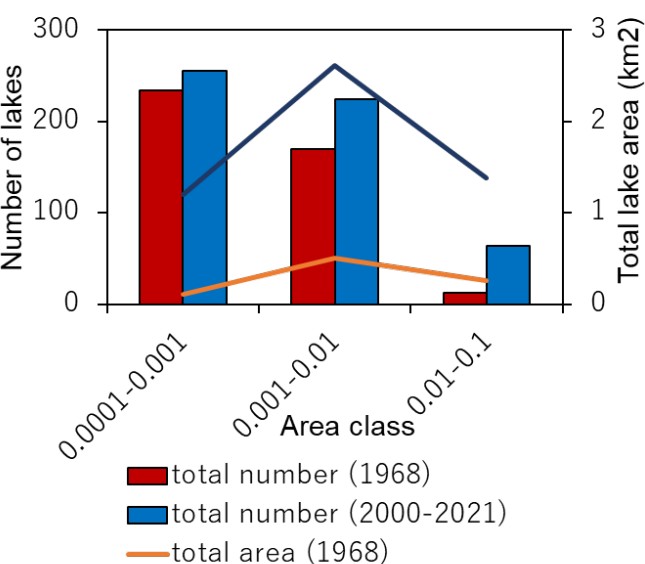

**Figure 3:** **Numbers and areas of the three area classes of glacial lakes in 1968 and 2000–2021.**

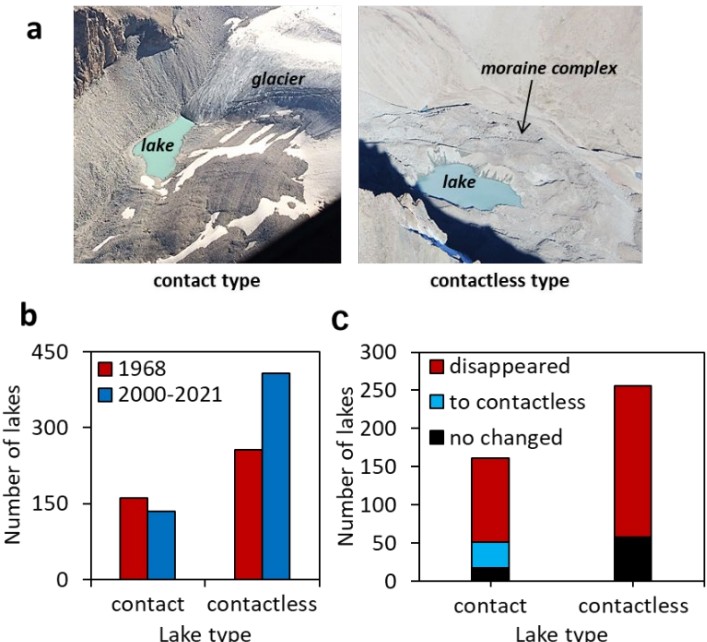

Figure 4: Contact- and contactless-type lakes. a) Examples of the two lake types. b) Number of glacial lakes according to type in 1968 and in 2000–2021. c) Changes in lake type from 1968 to 2000–2021.

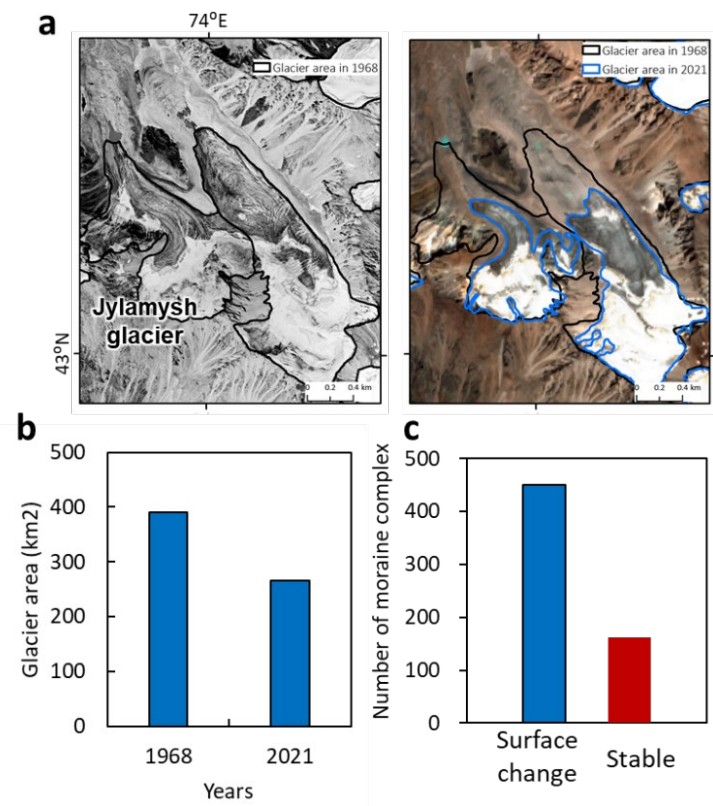

**Figure 5: Shrinkage of glaciers in the Kyrgyz Range using images from different years. a) Example of the Jylamysh Glacier.**
**CoronaKH-4Bimage from 1968 (left) and Planet Scope from in 2021 (right). b) Total glacier areas of the Kyrgyz Range in 1968 and 2021. c) The number of GMCs that showed a ground surface based on DInSAR data.**





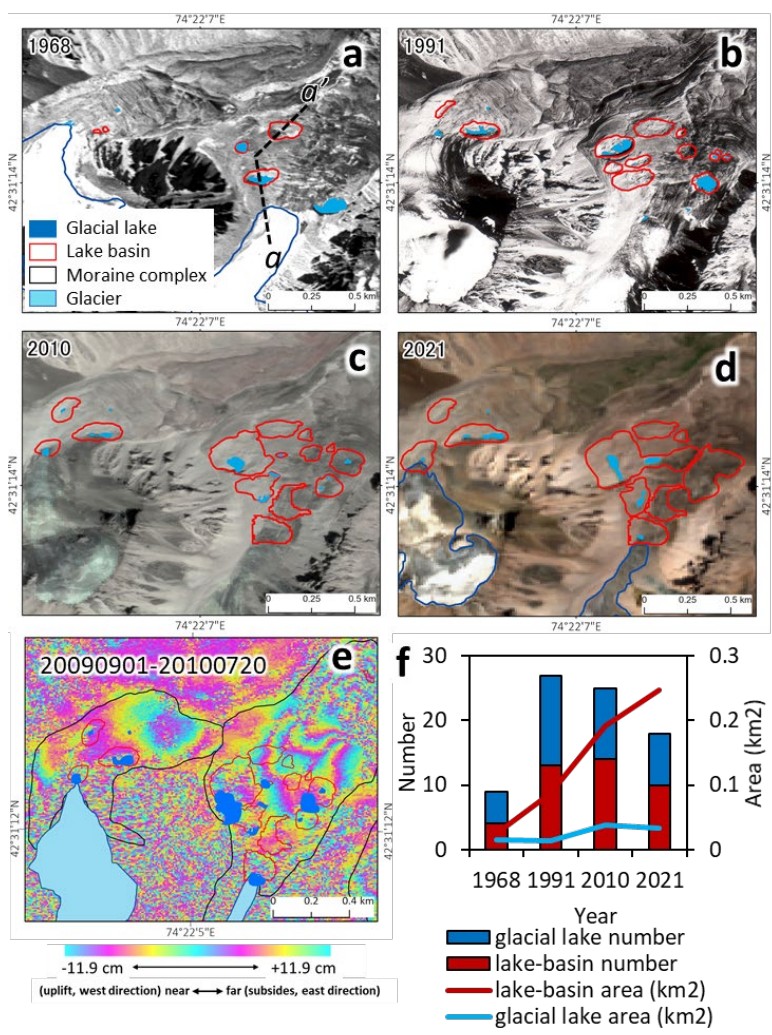

**Figure 6: Jylamysh GMC and surroundings. Location shown in Fig. 1. a) Orthorectified Corona image from 1968. Legend also applies to b-d. Dashed line marks profile used for Fig. 7c. b) Orthorectified AeroPhoto image from 1991. c) ALOS/PRISM-AVNIR-2 pansharpened image from 2010. d) PlanetScope image from 2021. e) Surface change area based on the DInSAR analysis in 2009–2010 using ALOS/PALSAR. f) Lake and basin numbers and areas based on images in a-d. Solid curves are the areas, with scale at right.**





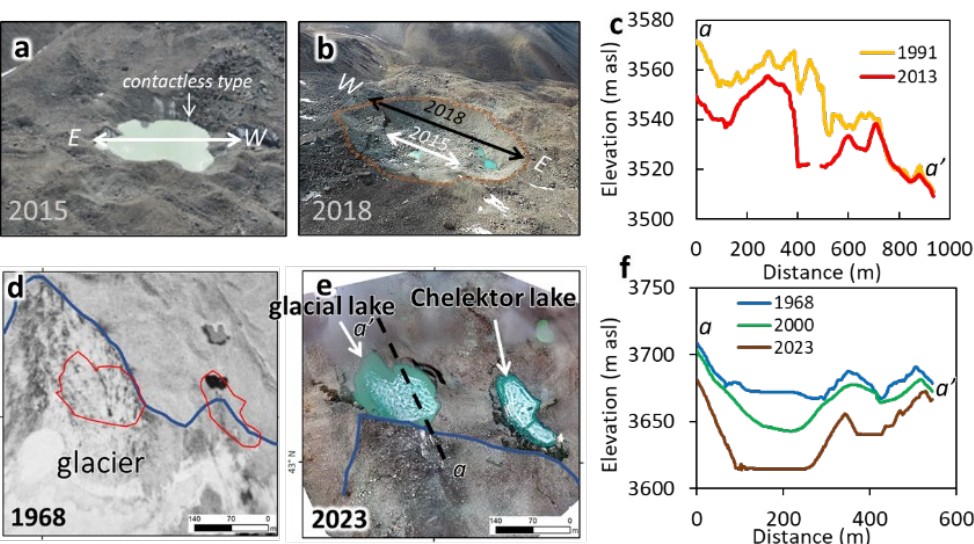

**Figure 7: Expansion of lake basins and newly formed glacial lakes near two glaciers in the central part of the Kyrgyz Range. Top row: near the Jylamysh glacier front. a-b) Change between 2015 and 2018 from High Mountain Asia aerial photographs. Red dotted line is lake basin perimeter, and the line arrows (white and black) marks the scale of expansion of the basin. c) Profiles along line marked in Fig. 6a. Bottom row: near the Chelektor glacier front. d) 1968 AeroPhotoDSM image. Red perimeters mark lake shores in 2023. e) Newly formed glacial lakes, image from UAV data. f) Profiles of lake basin marked in e).**

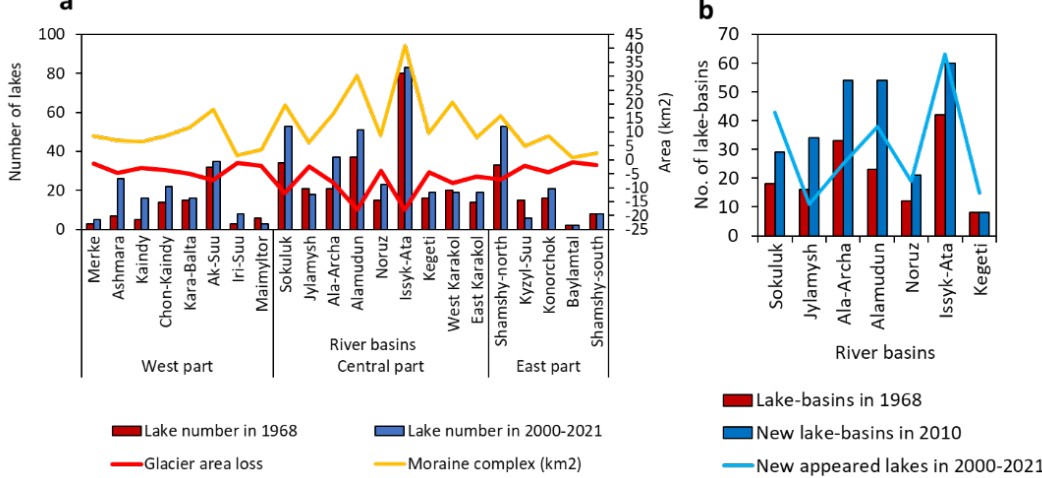

**Figure 8: Changes between 1968 and 2000–2021 for each river basin. a) Number of lakes, glacier areas, and moraine areas for all basins. b) Number of lake basins in 1968 and 2000-2021, and new lakes in 2000-2021 in the central part of the range.**



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
