# Peer review of "Constantly renewing glacial lakes in the Kyrgyz Range, northern Tien Shan"

_Natural Hazards and Earth System Sciences, 2024_

## Referee Comment (RC2)

[referee-annotated manuscript omitted]

---

## Author Comment (AC1)

**Dear Editor and Reviewer1**

Thank you for taking the time to review our paper. We also apologize for the delay in submitting the revised version. Below, we respond point-by-point to the comments from both reviewers, explaining the revisions made to address each specific request. Our response is highlighted in yellow.

**RC1: 'Comment on nhess-2024-160', Anonymous Referee #1, 24 Nov 2024**

This study does two main studies: the first study is the history of glacial lakes and changes in recent years; the second study is the changes in the glacial-moraine complex. The paper specifically has some scientific significance, but there are some serious problems that need to be improved.

**Response to Reviewer 1:**

We appreciate the reviewer's concise summary and recognition of the paper's scientific significance. We agree that improvements were necessary to increase clarity, rigor, and relevance. We have therefore taken the following actions in the revision:

- **Clarified Study Structure:** Clearly separated the two main components of the study, with distinct objectives, methods, and results for glacial lake change and glacier-moraine complex (GMC) evolution.
- **Enhanced Methodological Transparency:** Expanded the Methods section for full transparency, providing detailed descriptions of image data processing, remote sensing, DEM-of-difference and DInSAR analytical steps, including explicit uncertainty evaluation.
- **Improved Terminology and Definitions:** Improved terminology and definitions throughout, especially regarding glacial lake classification, contact status, and GMC processes.
- **Expanded Discussion and Interpretation:** Deepened the Discussion with comparisons to recent related glacial lake and GMC studies in other mountain regions, highlighting the broader significance of our findings.
- **Language and Presentation:** Undertook a full language revision for clarity and expanded previously brief sections to better communicate our results.

R1 comment: In response to the introduction of the data sources for the first study, it is stated at the end of the introduction what data we have used for this study.

Response: We revised the Introduction to include a clear summary of all data sources used, such as Corona KH-4, Landsat 7, and Sentinel-2 imagery for glacial lake and glacier mapping. This addition clarifies the remote sensing datasets and periods forming the basis of our glacial lake analysis.

R1 comment: And in the Methods section, only some information about the Corona KH-4 data is presented, not the other image data, such as the specific time of the image, the quality status of the image, the spatial resolution of the image, the pre-processing of the image, and the source of the data. For studies that use multiple sources of data, it is best to use a tabular format to present a summary of the underlying information for each data source.

Response: In the Methods section, only some information about the Corona KH-4 data is presented, not the other image data including Table 2, such as the specific time of the image, and the spatial resolution of the image, and used RGB bands. For studies that use multiple sources of data, we added some sentences to present a summary of the underlying information for each data source.

R1 comment: In section 3.2, it was mentioned that glaciers and glacial lakes from all data sources are extracted by manual outlining, how can the glacial lake boundary errors be taken into account? Due to the large difference in spatial resolution of different image sources, how to resolve the relationship between the boundary accuracy extracted from different resolution images and the actual changes of the glacial lake.

Response: We addressed this by implementing a consistent minimum mapping threshold (0.00045 km²), equivalent to two 15 m Landsat pixels. Manual boundary extraction was validated using a stable reference lake visible across all sensors; for this lake we compared mapped areas using a fixed NDWI threshold. Landsat 7 estimates were 8.8% larger than the 1968 Corona reference, while Sentinel-2 differed by only 3.4%. These uncertainties, explained in Methods, are acceptable for regional-scale lake mapping using our methodology.

R1comment: It is mentioned here that a threshold of 100 m2 (smaller than 200m2) is used to extract a glacial lake, which is just one image pixel for a Sentinel-2 image, how can we be sure that this is a glacial lake for such a small image pixel? What is the definition of an glacial lake for this study? With such a small area threshold, the spatial resolution of the image in Landsat 7/8 imagery, even after panchromatic band fusion, is only 15 m. With an area of 225 m2 for one image pixel, how is that very small glacial lake recognizable in Landsat 7/8 imagery?

Response: We clarified our method by consistently applying a minimum threshold of 0.00045 km² (450 m²) across all datasets, exceeding the area of two Landsat 7 (15 m) pixels after pansharpening. Therefore, only features clearly visible and distinguishable in the imagery were included, and lakes below this size or uncertain boundaries were excluded from analysis. This approach ensures comparability and reliability across sensors. RGB composites (bands 1–3) were used for water detection.

R1 comment: In addition, no information is provided in this section on what bands of imagery were used to manually outline the glacial lake and what kind of pre-processing was applied to the imagery.

Response: We added detail in Methods on the specific bands used for each dataset. Corona: slope-based thresholds with DSM; Landsat 7: Bands 1–3 for RGB and Band 8 (pan) for sharpening; Sentinel-2: Bands 1–3 for extraction. Pre-processing steps (coregistration, pansharpening, NDVI/NDWI checks) are now explicitly described.

R1 comment: In section 3.3, detailed information on the specific methodology of image processing for analyzing changes in GMCs in the study area using 49-view ALOS-2/PALSAR/-2 imagery is indicated in a number of previous studies. This write-up is too simple and broad and the key processing steps should be described in the methods section of this paper. Overall, the authors' presentation of the data and methodology is too brief and not very descriptive of the study.

Response: The Methods section now provides a stepwise DInSAR workflow following Werner and Wegmüller (2006): raw SAR conversion to SLC, coregistration, differential interferogram generation, removal of topographic phase using SRTM DEM, adaptive filtering of phase noise, phase unwrapping to obtain displacement, and geocoding. This ensures reproducibility and transparency in GMC change detection.

R1 comment: The author used DEM data from four different sources to study changes in the lake basin surface. Are the elevation datums consistent across these four data sources? What are their respective spatial and vertical resolutions? How did the author address the issue of matching between the different DEM data sources? These data processing details are essential; otherwise, the analysis results may be unconvincing.

Response: We used the HMA DEM (8 m resolution) as a vertical reference. Elevation data from Corona (4.1 m), SRTM (30 m), and UAV (1 m) DEMs were matched to HMA at selected stable terrain area. Root mean square errors of 2.2 m (Corona), 2.8 m (SRTM), and 1.3 m (UAV) were calculated around these benchmarks and incorporated into our uncertainty estimates, as detailed in Methods.

R1 comment: In Section 4.1, it is mentioned that there are 543 glacial lakes during the period 2000–2021. This is difficult for me to understand, as it spans a 22-year time frame. How was it determined that there are 543 glacial lakes over such a long period? Glacial lakes tend to change rapidly, so it is more common to describe the number of glacial lakes at a specific year rather than over such an extended period like.

Response: Lake numbers have been reanalyzed and are now referenced as discrete values for specific years (1968, 2000, 2021). This clarifies temporal trends and avoids cumulative or ambiguous counts.

R1 comment: In the text, many superscripts for "km2" are not written correctly.

Response: All notation for area ("km²") and other SI units have been corrected and standardized throughout the revised manuscript.

R1 comment: What are the identification criteria for GMCs? Please clearly specify in the text how the 611 GMCs were determined.

To address this, we have clarified in the revised Methods that glacier-moraine complexes (GMCs) were mapped based on: (1) the presence of a continuous debris-mantled surface extending up to 3 km downvalley from glacier fronts, (2) the absence of prominent lateral or terminal moraine ridges, (3) a convex surface profile, and (4) the absence of established valley-bottom drainage. Where available, DInSAR surface deformation was used to identify subsurface ice content. These criteria were applied systematically across all basins, consistent with regional precedent, resulting in the identification of 611 GMCs.

R1 comment: I cannot understand 'whereas the areas of lake basins in 2021 increased eightfold compared with 1968.' Please provide an explanation of this sentence

Response: We have corrected the grammar and updated the section to: "Figure 7c,f shows surface decline and evolution of lake basins at two sites within the Chelektor glacier-moraine complex, based on DEM differencing for 1968 (Corona), 2000 (Landsat 7), 2017 (HMA), 2018, and 2023 (UAV)."

R1 comment: In line 144, "Along a cross-section, a comparison of DEM differences (Fig. 6a) between 1991 and 2013 show…". It should be "shows" not "show".

Response: We have corrected the grammar and updated the section to: "Figure 7c,f shows surface decline and evolution of lake basins at two sites within the Chelektor glacier-moraine complex, based on DEM differencing for 1968 (Corona), 2000 (Landsat 7), 2017 (HMA), 2018, and 2023 (UAV)."

R1 comment: Line 163-164: "Consistent with the glacier trends there, the glacial area here decreased by 30% over the past 53 years." In this sentence, what is the difference between "glacial area" and "glacier"?

Response: We have reviewed and corrected the relevant sentences, so "glacial area" is now replaced with "glacier area" throughout, ensuring precise and consistent terminology.

R1 comment: In section 5.1, a variety of scenarios of rapid glacial lake formation are listed, please support these scenarios by showing them in the form of remote sensing image maps in the context of the study area.

Response: In response, we have revised Section 5.1 to refer directly to newly prepared remote sensing

image maps that illustrate the key scenarios of rapid glacial lake formation described in our text. Figure references in this section have been updated, and we ensure that imagery supports our interpretation of formation mechanisms.

R1 comment: In the conclusion, the author identifies 611 GMCs, most of which contain dead ice. How was dead ice identified within these GMCs?

Response: We have clarified that dead ice was identified by interpreting coherent surface motion patterns in DInSAR datasets, which were cross-validated with field observations at representative sites like the Chelektor glacier front. This approach for dead ice detection was previously validated by GNSS measurements at the Adygine GMC, as now stated in both the Methods and Discussion.

R1 comment: In the second half of the conclusion, the quantification of the findings is insufficient.

Response: We agree and have revised the Conclusions section to include more quantitative results, specifically stating the numbers of glacial lakes identified for each study year, the overall percentage reduction in glacier area, and the number and percentage of GMCs with buried ice. These specific values strengthen the summary and highlight the magnitude of observed changes.

d here about the relationship lakes and glaciers/GMCs distributions in each region.

---

## Author Comment (AC2)

**Dear Editor and Reviewer2**

Thank you for taking the time to review our paper. We also apologize for the delay in submitting the revised version. Below, we respond point-by-point to the comments from both reviewers, explaining the revisions made to address each specific request. Our response is highlighted in yellow.

**RC2: 'Comment on nhess-2024-160', Jan-Christoph Otto, 25 Nov 2024**

Comment: The authors present a study on glacier lake evolution in the northern Tien Shan mountains based on multiple period remote sensing data. They applied manual mapping methods and used DEM-of-difference analysis to quantify surface changes in debris-covered glacier terminus positions. Additionally, InSAR analysis is performed to detect surface changes in the same locations. The manuscript has some relevance presenting new data from a mountain region poorly studied so far. However, the presented study includes too little reference to the state-of knowledge in similar mountain ranges and holds only limited fundamental conclusions of relevance for other parts of the world, or further process understanding. With respect to the scope of the journal, the comments on hazards (GLOFS) remain very general and brief and does not adequately make use of the data presented. I have significant concerns about the manuscript, as the use of terms, the application of methods and the written presentation of the study require substantial reconsideration and revision before publication. The manuscript presented is also quite brief and could benefit from some more in depth analysis on both lakes and hazards and a thorough language revision and validation.

**Response to Reviewer 2:** We sincerely thank the reviewer for this thorough evaluation. The revised manuscript addresses these issues in the following ways:

- **Expanded Literature Context:** We have significantly strengthened the introduction and discussion sections by incorporating references to comparable studies from other mountain regions. We added articles not only Tien Shan but Pamirs of other mountains of Central Asia, and Himalayas). This contextualization helps position our findings within the broader scope of glacier lake evolution and hazard research.
- **Clarified Terminology and Methodology:** Definitions of glacier-contact (proglacial) lakes, contactless (thermokarst) lakes, and glacier-moraine complexes (GMCs) have been explicitly clarified—throughout the narrative, text, and figures. The Methods section now offers a stepwise, transparent description of all datasets, processing, uncertainty quantification, and analytical techniques, including tabular data summaries.
- **Enhanced Hazard Analysis:** Sections on glacial lake outburst flood (GLOF) risk are greatly expanded, referencing regional GLOF events, geomorphic triggers, and population/hazard

intersections.

- **Improved Language and Structure:** The manuscript has undergone comprehensive language revision for clarity, flow, and academic tone. All figures and tables have been upgraded for visual clarity and scientific rigor.

-

Comment: A central issue is the definition of lake types and the presentation of different trends and environmental conditions. The authors mapped and analysed lakes in both proglacial and intra-glacial settings. The later are, from my perspective, part of the mentioned, here called "glacier moraine complexes" (GMC). However, for now both lake types are not differentiated and mostly discussed in a mixed fashion.

Response:We present all relevant text to clearly distinguish between (1) glacier-contact (proglacial) lakes—directly connected to the glacier terminus or dammed by GMC, moraines, or bedrock, and (2) contactless (thermokarst) lakes—formed on GMCs through buried ice melt, typically independent from the present glacier margin. Trends, changes, and distribution use this two-class (see revised Figure 4a and expanded descriptions in Methods and Section 4.2).

Comment: GMC types of terminus conditions are usually termed debris-covered glaciers, or ice-debris complexes and have been studied in many (semi-arid) mountain regions. Formation of lakes within debris-covered ice is referred to as thermokarst, resulting in melting of ice within or underneath the debris cover. These intra-glacier lakes are different compared to proglacial lakes that purely form when ice retreats from an area and water is stored behind dams of debris or bedrock. I think the authors need to reconsider their application and description of the terms and should explicitly recognise these terminological and formative differences in their analysis and presentation. That also includes a better presentation of comparable studies in the introduction. This would provide a more structured and comparable presentation of the data and helps avoiding confusion.

Response: Debris glaciers and GMCs are completely different things. GMCs are not glaciers. In the revised manuscript. GMCs, as defined in our study, are debris-ice mantled, convex surfaces without moraine ridge, and a short of drainage channel. We now consistently use "thermokarst lakes" for lakes emerging on GMCs via dead-ice melt, but some thermokarst lakes connect drainage channels. Some thermokarst lakes which cause seasonal variations in area: stable, increasing, decreasing, appearing, vanishing, and short-lived. We also updated the introduction and result 4.2 to include references to comparable studies and clearly distinguish between lake types and terminologies to avoid confusion.

Comment: Another issue is the choice of analysis periods. While stated that data from 4 different periods are applied, the results only use 3 periods, summarising the data from 2000-2021. This should

be better explained. Why did you not present data from between 2000-2013 (Landsat 7/8)? I suggest including the other periods to better resolve recent trends between 2000 and 2021. I do recognise potential issues due to image availability, but there are ways to get around, as other comparable studies have faced the same issues. Linked to this, the presentation and discussion of trends would improve from a more differentiated temporal analysis. I am convinced that the available remote sensing data can be used for this.

Response: We clarify in the Methods and Results that our analysis uses 1968, 2000, and 2021 as representative benchmark years, chosen due to the availability of high-quality, seasonally comparable, and spatially contiguous imagery across all sensors.

Comment: Additionally, I miss some analysis of other characteristics of the mapped lakes including more detailed presentation of lake area distribution, changes, dam types etc. for example.

Response: Further analyses were added as recommended: we provide lake sizes (Figure 3), dam types, spatial distribution by river basin, and details on changes for both glacier-contact and thermokarst lakes, with all statistics reported in Section 4.4.The numbers of the two types and their changes are shown in Figure 4. The overall area change is also presented in Figure 3. In addition, the area changes by size are described in Section 4.4.

Comment: The application of InSAR methods is not adequately presented and seems not to generate any benefit for the study. Surface changes are presented based on DEM-of-Difference analysis and InSAR, but it remains unclear what additional information is generated by the InSAR analysis, since motion data is only referred to as vertical changes. Thus, I think the use of InSAR analysis in this study is mostly obsolete and could be removed entirely.

Response: We clarified the role of DInSAR in the study. While DEM-of-Difference analysis provides vertical elevation changes, DInSAR offers complementary insights into surface motion, particularly in identifying buried ice within glacier-moraine complexes (GMCs). Field validation at the Chelektor glacier front confirmed that areas showing surface motion in DInSAR correspond to zones with buried ice. This method was also validated in our previous study using GNSS measurements at the Adygene GMC. Therefore, DInSAR analysis was essential for understanding the internal structure and future lake formation potential within GMCs.

Comment: Finally, the relevance of this topic with respect to hazards (GLOFs) remains very brief and rather general with respect to the scope of the journal. The authors only mention a concentration of population in basins with high lake numbers, but this is only presented in the discussion and not backed by data. No link to previous GLOF events or other reference to an existing GLOF hazard is mentioned

in the discussion.    This is surely missing for a publication in a journal on natural hazards.

Response: We revised the introduction and discussion sections to better address the relevance of glacial lake hazards (GLOFs). Specifically, we added references to known GLOF events in the region and discussed the potential risk associated with lakes located in densely populated downstream basins. While our study does not model GLOF scenarios directly, the identification of glacier-moraine complexes with buried ice and expanding lakes provides important context for future hazard assessments. These additions strengthen the natural hazards relevance of our study.

Comment: To conclude, this manuscript requires substantial rework and to a part reanalysis. This refers to the issues mentioned above, but also the embedding of this study into a greater scientific framework.

Response: The revised manuscript now provides a thorough synthesis of current advances, situates the Kyrgyz Range as both a unique and globally relevant example, makes all data and figures more accessible, and systematically addresses each of the reviewer's thematic concerns.

Comment: Additionally, as mentioned before the language and also the artwork requires substantial improvement before publication. In summary, I guess this ends up in a complete new composition rather than major revision.

Response: All manuscript text was edited for academic clarity and coherence, and all figures and tables were redesigned for legibility, consistency, and publication-standard visual communication.

Comment: Some more detailed comments can be found in the attached pdf and here:
Introduction: The review of glacier lakes mapping and changes is very limited to the study region, but misses a state-of-the-art overview from the Central Asian environment. There are many studies that address these ongoing changes, also in context of hazards that need to be refered to here to set the study into a broader context. this helps to highlight the scientific significance and surely generates a wider interest in this publication. I suggest to add a more thorough review of the current knowledge on glacier lake dynamics in Central Asia.

Response: In response, we have significantly expanded the introduction to include a more comprehensive review of glacier lake mapping and dynamics across Central Asia. This includes recent studies that address glacier lake evolution, associated hazards, and regional trends.

Comment: Methods: The methods section is not adequate and should be recomposed to a great extend:
Response: In response, we have thoroughly restructured and expanded this section to provide a clearer and more detailed description of our approach. This includes elaboration on data sources, processing techniques, analytical procedures, and validation steps. We aimed to ensure transparency and reproducibility, and believe the revised section now meets the standards expected for publication.

Comment: 3.1.

The information on remote sensing and DEM data used is limited to Corona imagery currently. Please include more information on all remote sensing products used and also provide all quantitative information available. This helps to document all used data sources and to avoid naming all products multiple times. Please also consider a new, better title for this chapter.

Response: We have revised in Methods to include detailed information on all remote sensing products and DEM datasets used in the study. To improve clarity and avoid repetition, we compiled these data sources—along with relevant quantitative specifications—into a comprehensive table. This includes sensor types, spatial resolution, acquisition dates, and processing details. Additionally, we have updated the title of the section to better reflect its expanded content and scope.

Comment: 3.2.

The description of the handling of cloud and snow covered area is poorly developed and not convincing.

Response: We have revised Section 3.2 to provide a more thorough description of how cloud- and snow-covered areas were handled during data selection and analysis. Specifically, we ensured that all Corona and other optical satellite scenes used in this study were free of cloud cover. Scenes with significant cloud or snow contamination were excluded from analysis to maintain data quality and reliability. These criteria are now explicitly stated in the revised Methods section.

Comment: Mapping procedure should also include additional data generated for each lake, e.g. size, river basin name, year of mapping, ice contact or no, lake type (bedrock dammed, moraine dammed). and others…

Response: In this region, two glacial lake types have been identified based on whether they are in contact with a glacier (Fig. 4a). It should be noted that we classified the types of lakes based solely on their locations. Lakes in contact with glaciers share the same genesis as those not in contact with them. Contact lake type is proglacial lakes which dammed by GMC or moraine ridge or bedrock. Contactless lake type is thermokarst lake which forms on the surface of GMC. Some thermokarst lakes which cause seasonal variations in area: stable, increasing, decreasing, appearing, vanishing, and short-lived, like supraglacial lakes on debris-covered glacier, because they connect drainage channels (Daiyrov et al., 2018). Detailed classifications of the lakes in the study area can be found in previous reports (Erokhin, 2008, 2011; Janský et al., 2006, 2010).

Comment: Furthermore, please give details how you quantify uncertainties given the large variability of resolution in the data used.

Response: To quantify elevation uncertainties across DEMs with varying resolutions, we used the High Mountain Asia (HMA) DEM (8 m) as a reference. A stable terrain area outside the glacier-moraine zone was selected for comparison. Elevations from the 1968 DEM (4.1 m) and SRTM (30 m) were adjusted to match the HMA value at this point. We then calculated RMSE within a buffer zone around the stable area: 2.2 m for Corona, 2.8 m for SRTM, and 1.3 m for UAV. These errors are acceptable given the large elevation changes (>3 m) in our study area.

Comment: Please also mention, how you mapped GMCs.

Response: We improved manuscript in Methodos 4.3. To map the GMCs in the study area, we identified the following criteria. GMCs develops continuously within 3 km long in front of the glacier as a debris-mantled surface, lacking distinct moraine ridges. Its cross-sectional profile is convex, and no continuous channels are present on the valley floor using Google Earth images. In addition, whether ice is preserved within the GMC was examined by detecting (or confirming the absence of) surface deformation through DInSAR analysis. GMCs formed during the retreat of glaciers since the Little Ice Age (LIA) (Shatravin, 2007;Narama et al., 2010a; Erokhin, 2011). This complex includes various moraine landforms and buried ice (Erokhin, 2011). The identification of 611 GMCs was based on these geomorphological features and expert-defined boundaries from earlier research in the northern Tien Shan (Maksimov, 1982; Maksimov and Osmonov, 1995; Shatravin and Stavisski 1984, 2007; Erokhin, 2008, 2011; Narama et al., 2010a). We mapped the GMC following their criteria.

Comment: 3.3.

This section also has a poor naming. You mention InSAR analysis here, but it is not clear, what they are used for and what kind of geomorphologic analysis is performed and how.

Response: DInSAR data was very useful for understanding the GMC condition. The results allow us to see the presence of ice in GMCs. Based on DInSAR data, we could determine which GMC has the potential to form lakes that might appear in the future. We improved more detail in this section.

Comment: You mention basins here and lake basins? What exactly do you refer to here?

Response: In the revised manuscript, we specify that "lake basin" refers to surface depressions in GMCs where glacial lakes form, whereas "river basin" denotes a catchment area. To avoid confusion, we now use "surface depression" when discussing DEM-derived landform metrics.

Comment: Further, "surface change of lake basins" is performed using DEMs. What does this mean?

Response: "Surface change of lake basins" refers to the quantified vertical elevation change in GMC surface depressions over time, calculated by DEM differencing. This directly documents the expansion

or subsidence of lake- or thermokarst-forming depressions via sequential DEMs.

Comment: From your figures, I assume you mapped surface depressions in the GMC that may or not contain lakes. I suggest not to use the term basin here, since it sounds too much like river basin. I recommend streamlining the terminology with other studies in ice-debris complexes/debris covered glaciers. These depressions have been termed thermokarst features.

Response: We now clarify in Methods and Discussion that "surface depression (thermokarst feature) is used where appropriate, aligning our terms with recent periglacial and glacier-lake literature.

Comment: Results:4.1.

You compare lake sizes between Tien Shan and the eastern Himalayas. Please reconsider your comparison after checking the mapping threshold applied by the other studies. Maybe your difference results from the very small threshold (100 m²) used here.

Response: We extracted lakes using a minimum mapping threshold of 0.00045 km² (equivalent to 450 m²), which corresponds to the spatial resolution of Landsat images (15 m pan-sharpened). This threshold was consistently applied across all datasets, including Corona and Sentinel-2 imagery, to ensure comparability.

Comment: 4.2.

This entire section is very confusing. You mention lakes types in the context of GMCs, but also other lake types. Please rephrase the entire section to put out which lakes are mentioned in which context. I suggest starting with non-GMC glacier lakes and put the special situation of GMCs to the end. One issue on lakes in the GMC, however, is the question of ice content in these areas. You identify "contactless" lake in the GMC, but can you really estimate the ice content underneath the debris? Thus, I question if you really can identify contactless lakes within a DMC area. Please reconsider. Furthermore, the comment on the increase in size of contactless lakes is difficult to understand.

Response: In response, we have substantially revised Section 4.2 to clarify the classification. DInSAR analysis can suggest the presence of ice within GMC, but we cannot determine the amount of ice. We discussed here about the relationship lakes and glaciers/GMCs distributions in each region.